# Innovation in Cocoa Fermentation: Evidence from Patent Documents and Scientific Articles

**Luciana Lordelo Nascimento** [1], **Marizania Sena Pereira** [1], **Lorena Santos de Almeida** [1], **Larissa da Silveira Ferreira** [1], **Bruna Louise de Moura Pita** [1], **Carolina Oliveira de Souza** [1], **Camila Duarte Ferreira Ribeiro** [1,2] and **Alini Tinoco Fricks** [1,*]

1    Department of Bromatological Analysis, Graduate Program in Food Science (PGALI), Faculty of Pharmacy, Federal University of Bahia, Salvador 40170-115, Brazil; lucianalordelo@outlook.com (L.L.N.); marizaniasena@outlook.com (M.S.P.); llorena966@gmail.com (L.S.d.A.); lsferreira03@gmail.com (L.d.S.F.); brunalmpita@gmail.com (B.L.d.M.P.); carolods@ufba.br (C.O.d.S.); camiladuartef@ufba.br (C.D.F.R.)
2    Nutrition School, Federal University of Bahia, Salvador 40110-907, Brazil
*    Correspondence: alinitf@ufba.br

**Abstract:** This review aims to analyze the technological and scientific applications regarding cocoa fermentation through a prospective study of patent documents and research articles. The Espacenet database was used as a patent research tool by searching both the IPC code "A23G1" and the terms "cocoa" and "ferment*". A total of 130 documents were found—49 were related to the subject. The Scopus database was also searched for scientific articles using the terms "cocoa" and "fermentation". A total of 812 articles were found—517 were related to the subject. Cocoa fermentation has not yet reached technological maturity, despite the growth in patent documents and scientific research observed in the last two decades. The creation of the Cacao of Excellence Program (2009), among others, has incentivized sustainability and quality in cocoa-producing countries. Brazil, Colombia, and Indonesia are leading with scientific publications in the last 5 years, despite the lack of patents filed. The United Kingdom, France, China, Canada, and Germany, despite not being cocoa-producing countries, are the main holders of the technology. Patent documents analyzed relate to food science, biotechnology, engineering, and chemistry. Microbial biotechnology has gained attention as a key factor to produce a higher-quality cocoa bean. *Saccharomyces* is the most frequent genus of yeast used as a starter culture in patent documents. Some patent documents propose the addition of fruits during cocoa fermentation, but a few scientific studies have been found on the matter. Overall, technological applications and scientific studies have focused on improving cocoa quality. The cocoa market is expected to increase significantly in the next few years, representing an opportunity to develop high-quality cocoa using novel fermentation techniques.

**Keywords:** cocoa; intellectual property; microbial biotechnology; fermentation; microorganisms; inventions





## 1. Introduction

A patent is an exclusive right granted to develop an invention, which is a product or a process that provides a new technique or offers a new technical solution to a problem. Technical information about the invention must be disclosed to the public in a patent application, meaning that patents are granted by patent offices in exchange for a full disclosure of the invention. Therefore, the publication of patents facilitates the spread of new knowledge, accelerates innovation activities, and provides incentives and protection for individuals by offering them recognition for their creativity and the possibility of material reward for their inventions [1].

The global cocoa and chocolate market size was valued at USD 47.1 billion in 2021. The market is expected to reach USD 68.2 billion by 2030. Consumer demand for cocoa ingredients to develop premium-quality products is expected to be a factor driving market growth. The cocoa and chocolate markets have been expanding in response to an increased

awareness of the health benefits of cocoa in chocolate products [2]. The growth in the cocoa market has presented opportunities for businesses to expand and innovate and therefore to protect their innovations through patents. Among all the links in the cocoa value chain, the fermentation process is crucial to determining the quality and homogeneity of cocoa beans [3].

Cocoa bean fermentation is a biotechnological process in which microorganisms, enzymes, and environmental conditions play key roles in the development of flavor and aroma. The first stage of the process is characterized by exothermic alcoholic fermentation by the yeasts that are predominant due to the anaerobic environment, pH, and temperature conditions [3]. Yeast invertase enzymes convert the sucrose present in the pulp into fermentable monosaccharides, allowing the initiation of alcoholic fermentation, in which glucose is converted into ethanol and carbon dioxide. Furthermore, yeasts produce various aromatic precursor compounds, such as alcohols and esters, which positively contribute to the aromatic profile of the chocolate [4,5]. In addition, yeast pectinase enzymes, along with endogenous pectinase, are important to the degradation of the pectin in the pulp [6,7].

Simultaneous with the action of yeasts, lactic acid bacteria (LAB) consume sugars, generating lactic acid and other metabolites (acetic acid and ethanol, among others). LAB also consume citric acid, an organic acid present in the cocoa pulp and responsible for its low pH, which causes the pH of the fermenting mass to increase slightly [6]. LAB are also known to affect the sensorial properties and composition of chocolate [8]. The pulp consumption by yeasts creates a more aerobic condition that, together with the pH and temperature increase, favors the presence of acetic acid bacteria (AAB). AAB promote the exothermic oxidation of the ethanol produced by yeasts into acetic acid, causing a temperature increase in the cocoa mass and a decrease in pH. Under these conditions, acetic acid penetrates the cotyledon, promoting the death of the embryo and a succession of enzymatic and biochemical reactions that lead to the development of flavor and color [7,9].

Therefore, cocoa bean fermentation is of great technological and scientific interest in the cocoa value chain and inspires many opportunities for innovation that often require scientific research. The aim of this work is to provide a technological and scientific prospection of patent documents and scientific articles regarding cocoa fermentation, presenting consolidated data as well as scientific contributions to the technological evolution of cocoa fermentation.

## 2. Review Methodology

In order to analyze the current technology related to cocoa fermentation, patent documents were searched in the international database Espacenet in May 2023. Espacenet is a database of the European Patent Office (EPO) where inventions and technical developments from 105 countries can be accessed freely. The search strategy used terms (Cocoa, Ferment*) found in the titles and/or abstracts. In addition, one International Patent Classification (IPC) code (A23G1, which covers cocoa, cocoa products, and substitutes) was chosen to consider all patent documents related to the search objective. The Boolean operator (AND) was used to ensure that the code and terms were present in the documents evaluated. The truncation operator (*) was used to find derivations of the term Ferment*, such as fermentation and fermented.

The term "patent document" includes both submitted and granted patents. The search resulted in a total of 130 patent documents. After considering the titles and abstracts, 49 documents within the proposed theme were selected. The data regarding the available documents were exported to Microsoft 365 Excel for later analysis. Patent documents were analyzed by identifying the years of patent filing, country of origin, depositors, and area of application. Figures were generated using Tableau Public (2023 Tableau Software®) and Canva website.

Scientific articles were identified in the Scopus database in May 2023 by searching the terms "cocoa" AND "fermentation" in the titles, abstracts, or keywords. A total of 812 articles were found. After considering the titles and abstracts, 517 documents within the

proposed theme were selected. The data regarding the available documents were exported to Microsoft 365 Excel for later analysis. Scientific articles were analyzed by identifying the years of publication, country/countries of origin, authors, and area of application. Figures were generated using Tableau Public (2023 Tableau Software®) and Canva. Moreover, most of the relevant articles found were used in this review to provide a dialogue between patent documents and scientific research.

The results and discussion are divided into four parts: (a) the scientific and technological prospection (temporal evaluation, origin country), (b) the technological prospection (depositors, investors, and application area), (c) the scientific prospection (authors, subject area), and (d) trends in the global market.

### 3. Technological and Scientific Prospection: Timeline and Territoriality

*3.1. Temporal Evolution of Patent Documents and Scientific Articles*

The purpose of providing a temporal evolution is to analyze the development of articles and patents over the years. Figure 1 shows the patent documents and articles related to cocoa fermentation. In 1923, the first patent document was registered in the United States by Celia Mclaughlin in the state of New York—US1575372A [10]. This invention was related to a method of fermenting cocoa beans. The object of the invention was to produce a bland cocoa bean and to increase the output weight of fermented beans from a given weight of unfermented beans.

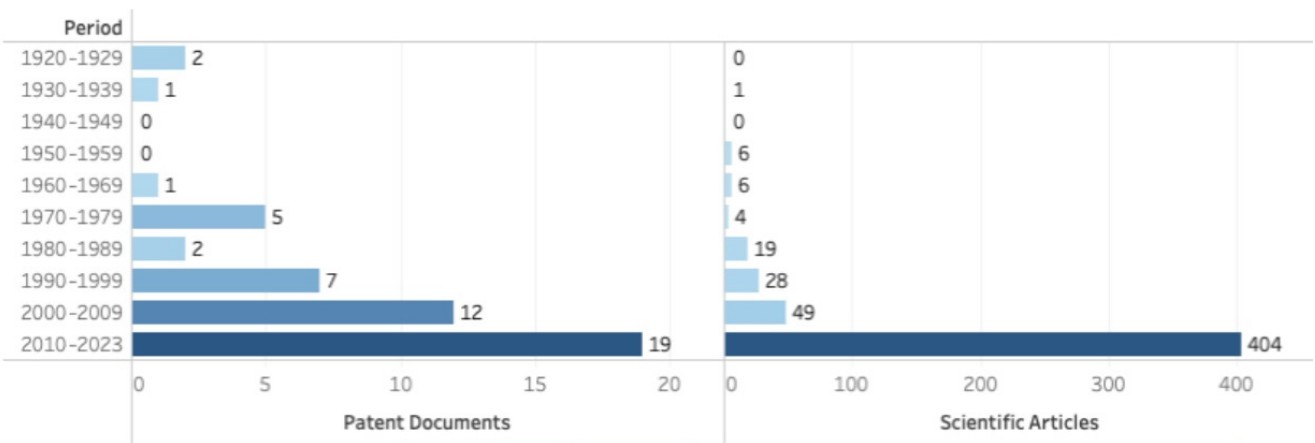

**Figure 1.** Temporal evolution of patent documents and scientific articles found in the Espacenet and Scopus databases regarding cocoa fermentation.

Up to 1999, a total of 19 patent documents were registered. These patent documents were related to food science (40%), engineering (30%), chemistry (15%), and biotechnology (15%). Since 2000, an increase in the registration of patent documents has been observed per year, with a total of 31 documents registered. A significant increase in technological applications related to biotechnology (38%) was also noticed, more specifically regarding the addition of microorganisms during fermentation. Moreover, a lack of patent deposits in 2023 was observed. It should be noted that publication can take place at various stages of the procedure. In many countries, patent applications are published 18 months from the filing date or the priority date [1], which may explain the lack of patents in 2023.

The first scientific article on the subject was published in 1930 in the Journal of Industrial and Engineering Chemistry, and it discussed the fermentation processes involved in the preparation of foods such as bread, tea, coffee, cocoa, and others [11]. Beginning in the 1980s, a growth in scientific articles was observed. This increase may have been encouraged by the creation of the International Cocoa Organization (ICCO) under the auspices of the United Nations in 1973. The first international agreement on cocoa was signed in 1973 in Geneva at a United Nations International Cocoa Conference, and this is a historical fact that symbolizes the importance of the cocoa market worldwide.

A significant increase in the number of publications began in 2010, with 78% of the articles published from that period onward. The present cocoa agreement was signed in 2010 and represented important breakthroughs such as the encouragement of cocoa quality by its members, the promotion of a sustainable cocoa economy, and the encouragement of research and the implementation of its findings. Since 2010, the creation of a variety of initiatives regarding sustainability and/or cocoa quality has been observed worldwide, such as the creation of the Cargill Cocoa Promise (2009), Mondelēz Cocoa Life (2012), Forum Nachhaltiger Kakao (2012), Cocoa Horizons (2015), and Forever Chocolate (2016), among others. Furthermore, the ICCO supported the creation of the Cacao of Excellence Program in 2009, which recognizes, preserves, and values cacao quality and flavor diversity.

Overall, there has been a progressive increase in the number of patent documents and scientific articles, which indicates that the technology, despite not being recent, has room for growth and development, not yet reaching scientific and technological maturity.

### 3.2. Country of Origin of Patent Documents and Scientific Articles

Patents are territorial rights. In general, the exclusive rights are only applicable in the country or region in which a patent has been filed and granted, in accordance with the law of that country or region. To protect an invention in several countries, applicants can submit patents according to the Patent Cooperation Treaty (PCT) through the World Intellectual Property Organization (WIPO). The Patent Cooperation Treaty (PCT) assists applicants in seeking international patent protection for their inventions, helps patent offices with their patent-granting decisions, and facilitates public access to a wealth of technical information relating to those inventions [12].

Europe is the continent that holds the largest amount of patent applications (36.73%), followed by North America (6.12%) and Asia (5.2%). The United Kingdom (12.2%), France (8.16%), China (6.12%), Canada (4.08%), and Germany (4.08%), despite not being cocoa-producing countries, are the main holders of the technology (Figure 2).

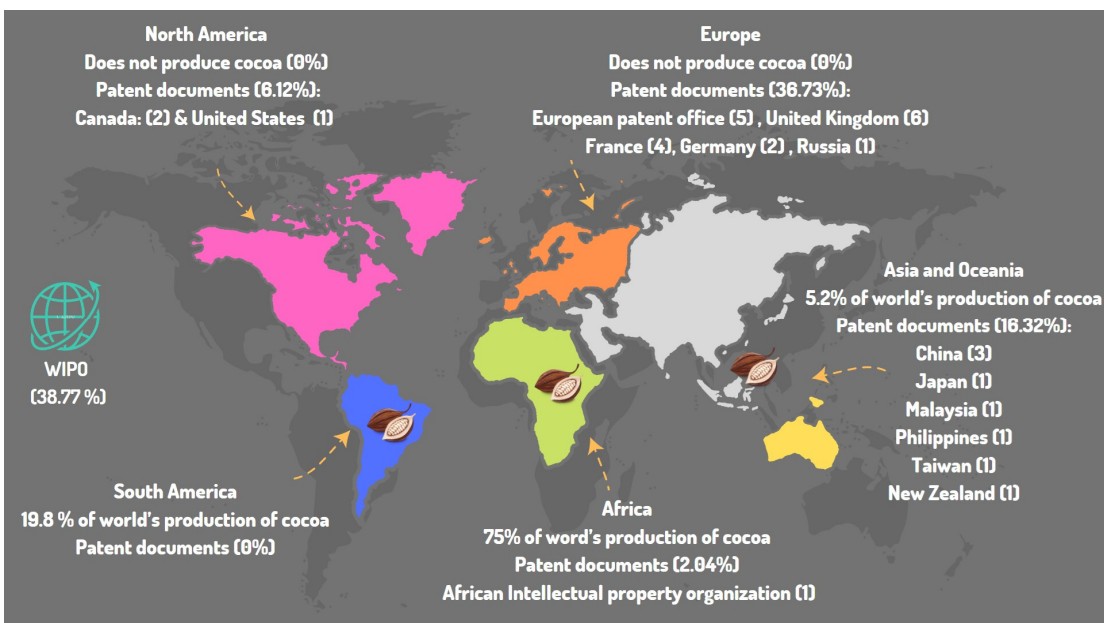

**Figure 2.** Worldwide distributions of patent documents found in Espacenet regarding cocoa fermentation and cocoa-producing regions.

Africa is responsible for around 75% of cocoa production worldwide [13], but only one patent has been filed through the African Intellectual Property Organization by Nestlé [14]. Brazil, the largest producer in America [13], has no patents filed. Although ahead of other emerging economies, Brazil demonstrates less significant relative results in international patents than in publications [15]. Malaysia is the only cocoa-producing country that

owns a territorial patent—MY106273A [16]. It is important to highlight that 38.77% of patent documents regarding cocoa fermentation have been filed through the WIPO and are therefore protected simultaneously in several countries.

Brazil leads by far in scientific publications regarding cocoa fermentation, followed by Germany and Belgium. Brazil is the seventh-largest cocoa producer in the world and the largest producer on the American continent, according to the International Cocoa Organization [13]. Despite its large production, it was historically known as a bulk (or conventional) cocoa producer and was not on the ICCO list of fine or flavor cocoa-producing countries. The ICCO Ad Hoc Panel list of fine flavor cocoa-exporting countries was revised by the Ad Hoc Panel on Fine Flavor Cocoa in 2004, 2008, 2010, 2015, and 2019. For example, in 2010, the ICCO panel noted that as of yet, there were no fine cocoa exports from Brazil. However, the panel declared Brazil as a country with the potential to produce and export fine or flavor cocoa in the near future [17]. Also, in 2010, the Cocoa of Excellence competition took place at the Salon du Chocolat of Paris. Of the 148 cocoa samples received and analyzed, 50 made it through the Cocoa of Excellence selection process and competed in their geographic areas of origin: West Africa, Central America and the Caribbean, South America, or Southeast Asia and the Pacific. That year, a Brazilian producer received a Cocoa of Excellence Award. The following year, he was awarded it again. An updated list of fine flavor cocoa-exporting countries was approved by the ICCO panel in 2019 and included Brazil as a fine cocoa exporter [18]. The international recognition of Brazil as a producer of high-quality cocoa beans may represent a milestone in the increase in scientific research worldwide (Figure 2) and in the country.

In the last five years, besides Brazil, Colombia and Indonesia have led in scientific production, which emphasizes that cocoa-producing countries are now more interested in studying the fermentation of cocoa beans. However, it is notable that these countries, despite their scientific leadership, do not own patented cocoa fermentation technologies.

## 4. Technological Prospection

### 4.1. Main Depositors and Key Investors of Patent Documents

Applicants for patents can be classified as private companies, universities, research institutes, or independent inventors. Regarding the main patent applicants, independent inventors dominate the sector (Figure 3), with 48.9% of the patent documents, followed by private companies (44.3%), universities (4.6%), and research institutes (1%). Despite the predominance of independent inventors, private companies individually hold the largest number of patents.

Universities and research institutes have been responsible for the notable growth in scientific research on cocoa fermentation in the last decade. However, this is not reflected in commercialization, as they have not shown a strong ability to convert research into innovation. Only five patents were submitted by universities and research centers located in Europe and Asia (Figure 3). To overcome this, some authors have stated that the creation of technology transfer offices within universities can support the commercialization of research [19]. Universities can also outsource their research to the established technology transfer office of another university, and that may be an opportunity for academics to engage in research commercialization and increase the number of new discoveries and provisional patent filings [20]. Besides that, the direct industrial funding of university research may be the best mechanism for maximizing return on research investment [21]. Therefore, partnerships between universities and the private sector are fundamental to boosting the participation of academics and researchers in cocoa-related innovation.

Nestlé SA is the company with the highest number of patent documents, holding 5 of the 49 patents assessed. Its patents are related to depulping cocoa beans [14] to enhance the quality of fermented beans and to providing an enzymatic treatment using proteases [22,23] to improve the composition of aroma precursors. Mars Inc., based in the United States, is the world's leading manufacturer of chocolate and holds four patent documents on cocoa fermentation. For example, patent document WO2014127130A1 [24] relates to methods of

processing cocoa beans under anaerobic conditions at a controlled temperature with the addition of a starter culture.

The French company Barry Callebaut submitted four patents related to cocoa beans. Nicholas Camu was the most frequent inventor in Barry Callebaut's filings and overall, with three documents related to the addition of starter cultures [25,26] or fermented pulp during the fermentation of cocoa beans [27]. He also developed scientific studies related to the dynamics of bacteria [28,29] and yeast [30] involved in the spontaneous fermentation of cocoa beans. Other companies such as Ritter, Natraceutical, and Golden Hope Plantations Berhad are also the main holders of patents.

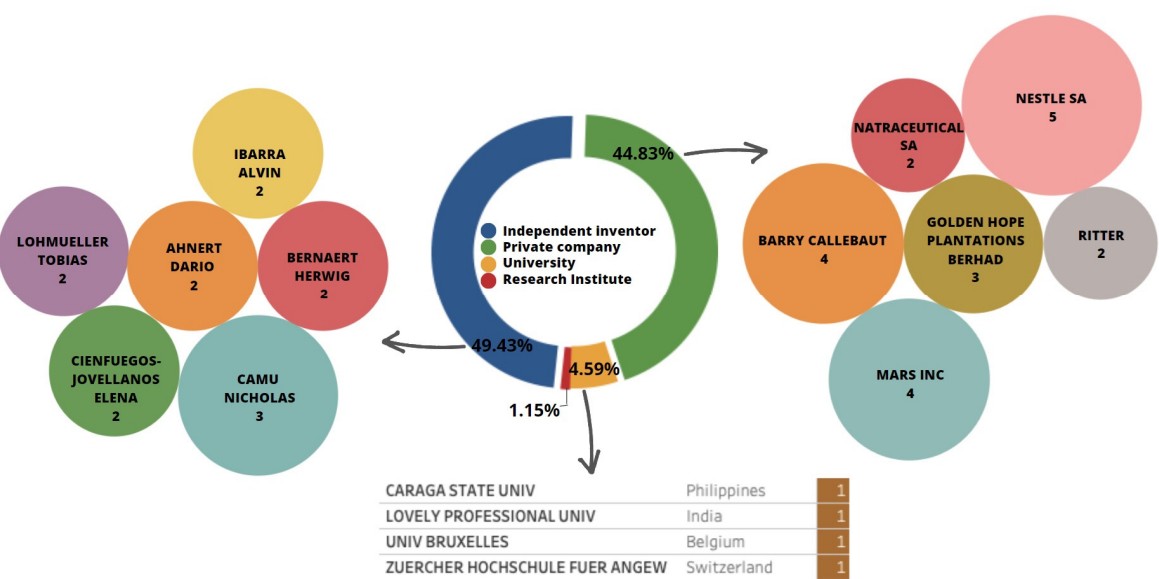

**Figure 3.** The main players and types of depositors of patent documents found in the Espacenet database related to cocoa fermentation.

### 4.2. Technological Domains of Patent Documents

Cocoa fermentation is a biotechnological process in which microorganisms, enzymes, and external conditions such as temperature, pH, and oxygen concentration play a key role in the development of flavor and aroma. Therefore, it is a process with great potential for technological exploration. Figure 4 illustrates the main technological domains related to the patent documents. Food science (15) and biotechnology (13) represented the greatest contributions, followed by engineering (12) and chemistry (9).

#### 4.2.1. Food Science

Of the 15 patents related to food science, 7 concern the addition of substances during fermentation, including edible plants (1), sugar-containing solutions (4), and ethanol (2). Previous analysis suggests that the cocoa beans absorb the flavors from the aromatic pulps of fruit during fermentation [27]. Ahrert Eskes is the main applicant in patent document WO2009103137A2 [31], which proposes a process to obtain fermented cocoa beans with modified flavors through the addition of aromatic substances such as fruits and pulps during the cocoa bean fermentation process. Positive changes in acidity, sugars, and acids were observed in cocoa beans of the CCN-51 variety, traditionally known as bulk cocoa, fermented with probiotic microorganisms, passion fruit (*Passiflora edulis*), and banana (*Musa paradisiaca*) at a laboratory scale for six days [32]. The substitution of cocoa pulp with an artificial cocoa pulp medium during cocoa bean fermentation has also been used to study various fermentation conditions that may improve the quality of cocoa beans [33]. Regarding the organic acids that diffuse into the cotyledons, the timing of initial entry and pH are crucial for optimum flavor development [34]. Patents related to food science also include the depulping of cocoa beans prior to fermentation as a strategy to reduce

the acidity of fermented cocoa beans. Some studies have shown that the partial removal of pulp before fermentation has resulted in better yield and quality without disturbing successful fermentation and does not affect the flavor development [7,35].

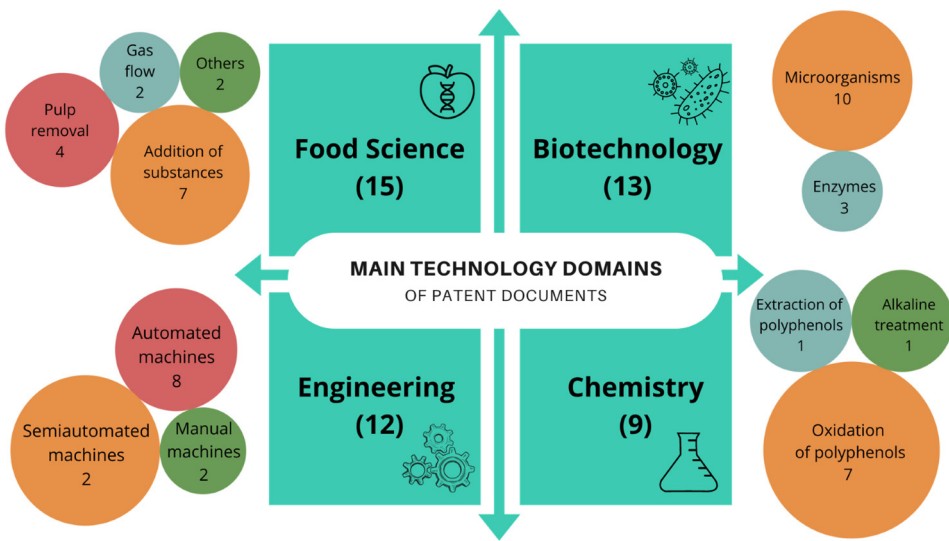

**Figure 4.** Main classification of patent documents found in the Espacenet database regarding cocoa fermentation according to their technological domains.

4.2.2. Biotechnology

Patent documents regarding biotechnology are mostly related to the addition of microorganisms during the fermentation process as a strategy to improve cocoa quality and maintain reproducibility between fermentation batches, with 11 documents assessed. Figure 5 shows the main microorganisms claimed in patent documents. Many patent documents propose the addition of a consortium of yeast and bacteria as a starter culture during the fermentation process.

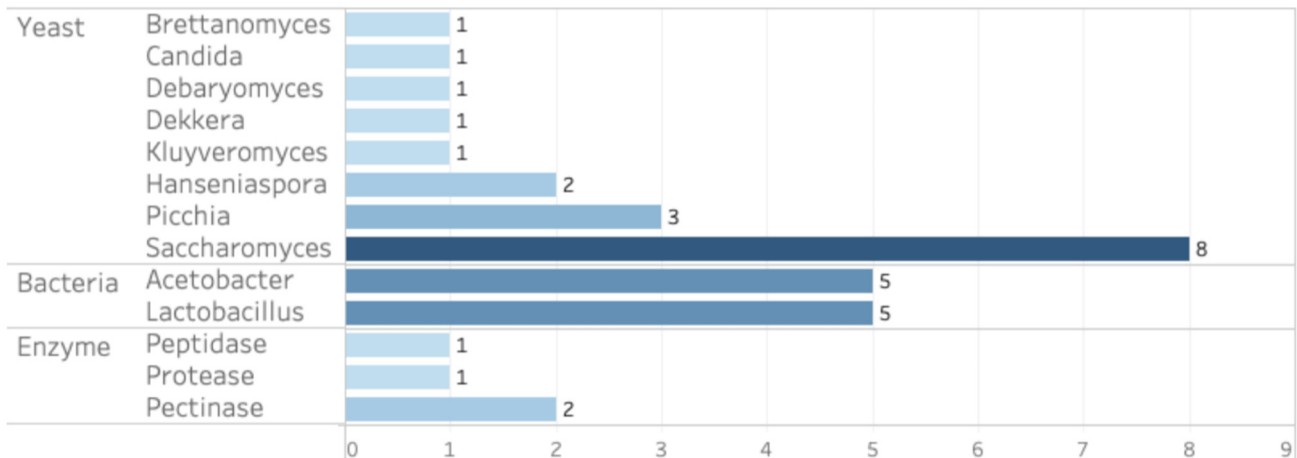

**Figure 5.** Microorganisms (yeast and bacteria) and enzymes claimed by patent documents found in the Espacenet database regarding cocoa fermentation. The colored horizontal bars correspond to the number of patent documents claiming the microorganism/enzyme.

*Saccharomyces* is the most frequent genus of yeast used as a starter culture in patent documents, followed by *Pichia* and *Hanseniaspora* (Figure 5). *Saccharomyces cerevisiae* is the yeast most frequently found in cocoa fermentation and is proposed as a starter culture in the scientific articles assessed. It has been reported to ferment faster [36], and a wide diversity of volatile organic compounds responsible for aroma are produced when using

*S. cerevisiae* and *P. kudriavzevii* [36,37]. Several studies have also reported on the pectinolytic activity of various yeast species such as *Saccharomyces cerevisiae*, *Pichia kudriavzevii*, *Candida norvegensis*, and *Kluyveromyces marxianus* during cocoa fermentation [38]. This activity degrades the viscous pectin mucilage of the pulp and allows the later penetration of acetic acid into the cotyledon.

Patent document EP3837988A1 [39] proposes the improvement of cocoa bean quality through the preparation of a cocoa honey–natural leaf-based inoculant. The patent claims a process in which cocoa honey is separated from the beans to prepare an inoculum containing a triturated leaf naturally rich in yeast, such as *Musa acuminata* or *Musa balbisiana*. It represents a technological application that is basically free of cost.

In addition to the patents involving the use of microorganisms during cocoa fermentation, four patents are related to the addition of enzymes such as pectinase (2), peptidase (1), and protease (1) during the fermentation process. Overall, the higher number of microorganism-related patents may be due to the cost of enzymes, and their instability and non-reusability limit their applications. This barrier can be overcome by the immobilization of enzymes, which reduces the costs, increases their stability, and thus is widely pursued for efficient and environmentally friendly catalysis [40]. Patent document CN106417844A [41] relates to the addition of 0.5% pectinase at 40 °C for 90 min as a strategy to enhance the enzymatic hydrolysis of pectin and accelerate the fermentation process.

### 4.2.3. Chemistry

Nine patent documents relate to chemistry and describe the extraction of polyphenols from unfermented or partially fermented cocoa beans (7), the oxidation of polyphenols to reduce the astringency of cocoa beans (1), and the use of an alkaline treatment for free fatty acid reduction (1).

Cocoa is a rich source of phenolic compounds, containing approximately 6% polyphenols by dry weight in the Forastero variety. Epicatechin is the main monomeric flavanol in cocoa, accounting for approximately 35% of the total phenolic content [42]. Polyphenols are responsible for the sensation of astringency and bitterness in cocoa and chocolate [43]. During cocoa processing, the total phenol content is reduced by the presence of enzymes that promote the oxidation of polyphenols [44], with the greatest losses occurring in the fermentation and drying stages, where more than 70% of the epicatechin content is reduced [34,42]. Therefore, there is a technological interest in promoting the oxidation of polyphenols as a strategy to reduce the astringency of cocoa and improve consumer acceptability. Patent document WO0022935A1 [45] proposes the production of cocoa with a standardized flavor through a two-step process. In the first step, the unfermented beans are treated with acetic acid to destroy the cellular and subcellular structures, and in the second step, they are ground and subjected to an oxidation treatment.

On the other hand, the extraction of polyphenols from cocoa beans has a therapeutic purpose. It is known that polyphenols are antioxidants that play a main role in the prevention of chronic diseases caused by oxidative stress, such as cancer, cardiovascular diseases, and diabetes. Indeed, cocoa and its flavanols can interfere in the initiation and progression of these diseases through different mechanisms [46]. Therefore, the extraction of polyphenols from unfermented or partially fermented cocoa beans has attracted attention as a technological application. For example, patent document FR2812873A1 [47] proposes a method for the extraction of fermented or unfermented cocoa beans using solvents such as methanol, ethanol, acetone, 2-butanol, and 2-propanol 2, followed by filtration and the distillation of the filtrate to obtain an extract rich in polyphenols. The extract contains 15–65% polyphenol content, with antioxidant and radical scavenging activity useful in food, cosmetic, or drug applications. JP2010187704A [48] proposes a method for commercially producing cocoa polyphenols from cocoa beans for use as food, food intake, or pharmaceuticals using a microwave-mediated extraction or solvent.

### 4.2.4. Engineering

Engineering-related patent documents are associated with automated or semiautomated machines or apparatus for fermenting cocoa beans to achieve better control of the processing parameters and reduce hand labor.

Patent document WO2021170197A1 [49] proposes an automated revolving cocoa fermenter with temperature monitoring. The machine has a container in the shape of a horizontal cylinder with a draining system that is triggered to turn when the fermentation temperature reaches a certain value. Revolving cocoa beans provides aeration to the mass and ensures the predominance of AAB, which oxidizes ethanol to acetic acid in an exothermic reaction that maintains the cocoa mass at temperatures up to 49 °C [3,50]. The acetic acid product penetrates the cotyledon but is very highly volatilized during the drying step that succeeds fermentation and therefore does not negatively impact cocoa acidity. Temperature drops in the middle of fermentation may indicate that AAB is losing predominance to LAB. LAB produces lactic acid that is not easily volatilized during drying and results in more acidic cocoa [51]. Therefore, revolving cocoa beans is a strategy to produce higher-quality cocoa [52,53]. In addition, the rounded shape of the fermenter allows for better heat transfer during fermentation and contributes to obtaining a standardized bean. Overall, fermentation methods impact the physical and chemical characteristics of cocoa [54], as they affect the microbial population during fermentation [55,56].

## 5. Scientific Prospection

### 5.1. Authors and Universities

The top 10 authors presented in Figure 6 are responsible for 30.3% of the publications and are from Belgium (49), Germany (39), Brazil (32), Ghana (14), and France (13). Luc De Vuyst (33) from Belgium and Rosane Schawn (22) from Brazil lead in journal publications on cocoa fermentation. Nicholas Camu is the only author in the top 10 list affiliated with a private company, Barry Callebaut. He is also the depositor of patent documents WO2011012680A2 [25], WO2009138419A2 [27], and WO2007031186A1 [26], together with Barry Callebaut.

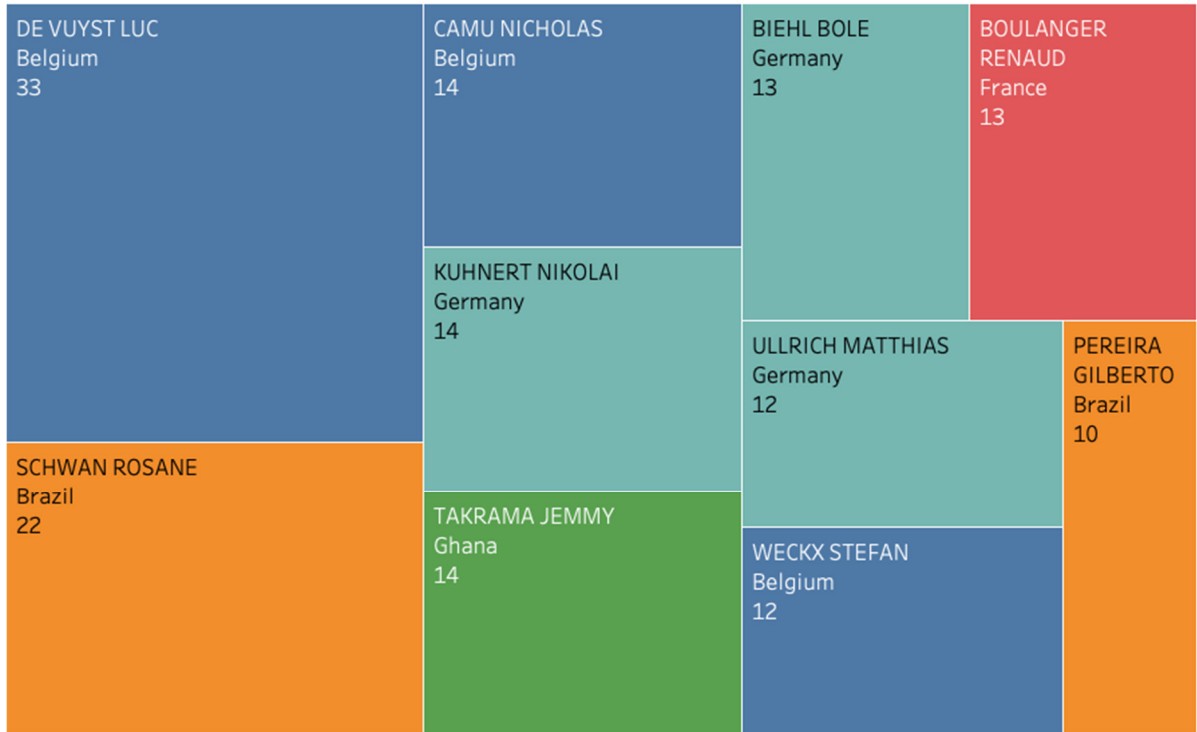

**Figure 6.** The top 10 authors with the highest number of publications on cocoa fermentation in the Scopus database.

The top 10 universities and research institutes that publish the most are in Belgium (1), Brazil (3), Germany (1), Ghana (1), and France (3), in accordance with the above-affiliated authors, as well as Malaysia (1). The university that publishes the most is Universidade Estadual de Santa Cruz (UESC) in Brazil, even though none of the top 10 authors are affiliated with UESC. In the last five years, new authors have composed the main authors list. Among them, Marcelo Franco, Ana Paula Uetanabaro, Rachel Rezende, and Carla Romano are affiliated with UESC and therefore contribute to UESC's number one university ranking. UESC is in Ilhéus, Bahia, a Brazilian region recognized as being a strong cocoa production center. The creation of the Cocoa Innovation Center (CIC) in 2017 within the UESC facilities has probably contributed to the number of its publications. The CIC aims to build, consolidate, and disseminate knowledge about cocoa and quality chocolate, with a simultaneous focus on applied research and bringing buyers closer to quality cocoa producers [57]. Furthermore, the appearance of Universiti Putra Malaysia among the top 10 universities can be explained by the contribution of many authors with a smaller number of published articles.

### 5.2. Subject Area of Scientific Articles

As shown in Figure 7, most studies have been in the agricultural and biological sciences (35%); biochemistry, genetics, and molecular biology (11%); and immunology and microbiology (10%). Table 1 lists the main purposes of articles related to cocoa fermentation in the last five years. Overall, the purpose of these articles is related to increasing the quality of cocoa beans through various approaches, such as the use of starter cultures, the study of potential microorganisms through genomics and metagenomics, and the use of enzymes, fruits, or techniques to optimize fermentation methods and conditions. Besides that, some articles consider the establishment of methods for measuring the fermentation index, while others propose the characterization of cocoa by determining antioxidant potential, phenolic content, and volatile and/or non-volatile compounds. These findings are in accordance with the global cocoa agreement signed in 2010 that encourages the promotion of cocoa quality and research.

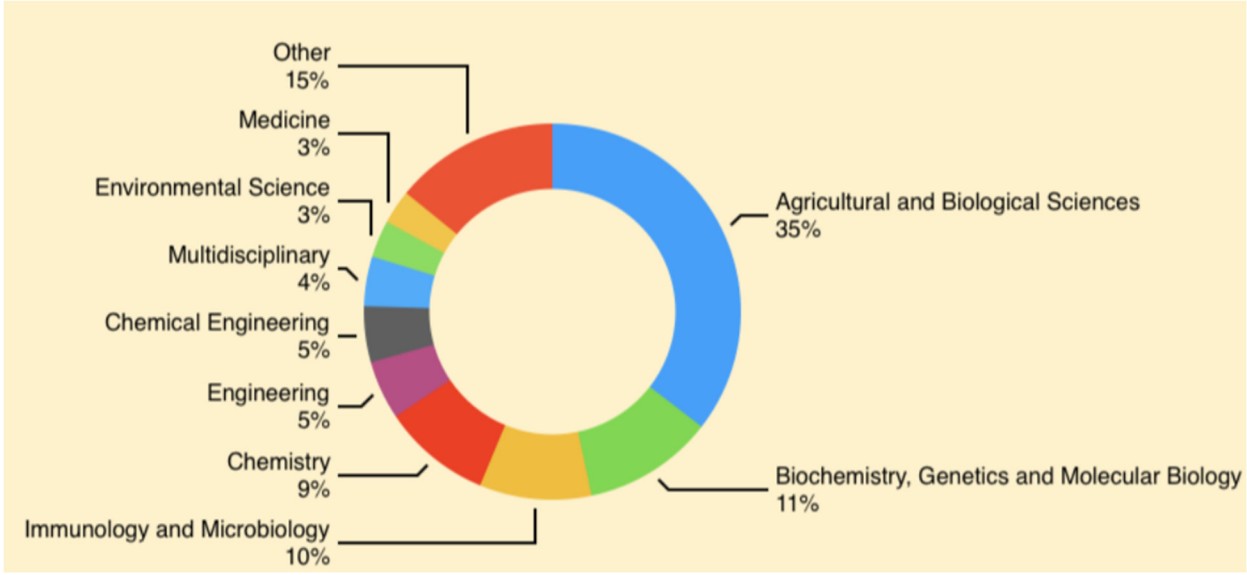

**Figure 7.** Application areas of published articles on cocoa fermentation in the Scopus database.

Given the important role of microorganisms in cocoa fermentation, several studies propose the use of starter cultures as a strategy to improve cocoa quality and maintain reproducibility among fermentation batches [8,58–60]. Potential starter cultures have also been studied to elucidate microbial diversity, structure, and functional and metabolic pathways. Some authors state that microbial cocoa fermentation inoculant composition

should also consider the region where fermentation is performed [61]. Table 2 shows the main articles in the last five years related to the use of starter cultures in cocoa fermentation.

**Table 1.** Main purposes of articles published on cocoa fermentation in the last 5 years.

| Purpose | How | References |
|---|---|---|
| To increase cocoa quality during fermentation | Using starter cultures | [9,37,58–60,62–75] |
| | Using enzymes | [76–78] |
| | Using fruits and probiotics | [32] |
| | By changing/optimizing fermentation method or conditions | [52–54,79–81] |
| | By studying potential starter cultures (e.g., metagenomics) | [61,82–89] |
| To measure fermentation quality/determine origin | Spectroscopy techniques | [90,91] |
| | Terahertz hyperspectral imaging | [92] |
| | Machine-learning techniques | [93,94] |
| To characterize cocoa beans during fermentation | Measurement of phenolic content, antioxidant activity, and volatile and non-volatile compounds | [5,95–114] |

**Table 2.** Main articles that investigate the use of starter cultures in cocoa fermentation in the last 5 years.

| Title | Microorganism | Purpose | Findings | Reference |
|---|---|---|---|---|
| An in-depth multiphasic analysis of the chocolate production chain, from bean to bar, demonstrates the superiority of *Saccharomyces cerevisiae* over *Hanseniaspora opuntiae* as functional starter culture during cocoa fermentation | *Saccharomyces cerevisiae*, *Hanseniaspora opuntiae* | To examine the effects of the cocoa isolate *H. opuntiae* IMDO 040108 as part of three different starter culture mixtures compared with spontaneous fermentation | The inoculated *H. opuntiae* strain was unable to prevail over background yeasts present in the fermenting cocoa mass. Cocoa fermentation processes inoculated with a *Saccharomyces cerevisiae* strain enhanced flavor production during fermentation, reflected in richer and more reproducible aroma profiles of the cocoa liquors and chocolates. | [65] |
| Effect of microencapsulated inoculum of *Pichia kudriavzevii* on the fermentation and sensory quality of cacao CCN51 genotype | *Pichia kudriavzevii* microencapsulated | To establish the effective dose for applying the microencapsulated yeast *Pichia kudriavzevii* as a microbiological starter of fermentation for the cocoa variety CCN-51 | A 2% microencapsulation of *Pichia kudriavzevii* is suitable for cocoa starters. The higher presence of volatile compounds such as 2,3-butanediol, associated with cocoa aroma, and 1-phenyl-2-ethanol and acetophenone, associated with aromatic descriptors of fruity and floral series, was observed. | [64] |
| Fine Cocoa Fermentation with Selected Lactic Acid Bacteria: Fermentation Performance and Impact on Chocolate Composition and Sensory Properties | *Lactiplantibacillus fabifermentans* *Furfurilactibacillus rossiae* | To evaluate the impact of adjunct cultures of selected lactic acid bacteria (LAB) on the fermentation parameters, chemical composition, and sensory profile of fine cocoa and chocolate | The addition of adjunct cultures influenced the proteolytic processes and the free amino acid profile and increased the complexity of the flavor profile of the chocolate. | [8] |

**Table 2.** *Cont.*

| Title | Microorganism | Purpose | Findings | Reference |
|---|---|---|---|---|
| Ecology and population dynamics of yeast starter cultures in cocoa beans fermentation | *Saccharomyces* spp., *Kloeckera* spp., *Candida* spp., and *Rhodotorula* spp. | To understand how microorganisms and their proliferation influence cocoa bean fermentation | *Saccharomyces* ssp. and *Hanseniaspora* ssp. strains can be used together as starter cultures in cocoa bean fermentation. | [67] |
| *Saccharomyces cerevisiae* and *Pichia manshurica* from Amazonian biome affect the parameters of quality and aromatic profile of fermented and dried cocoa beans | *Saccharomyces cerevisiae*, *Pichia manshurica* | To evaluate the physical and chemical transformations of cocoa beans during fermentation after inoculation with starter cultures of yeast species *Pichia manshurica* and *Saccharomyces cerevisiae* | The use of *Pichia manshurica* presented a superior performance compared to that of the culture of *S. cerevisisae* with respect to the quantity of desirable aromatic compounds for the production of chocolate, low acidity level, and high content of phenolic compounds. | [58] |

## 6. Trends

The world cocoa market distinguishes between two broad categories of cocoa beans: "fine" cocoa beans and "bulk" cocoa beans. The ICCO (2018) defines fine cocoa as a cocoa that is free of defects while providing a complex flavor profile that reflects the expertise of the producer and the "terroir", or sense of the particular environment where the cacao is grown, fermented, and dried. Judging from this generic definition, the concept of fine cocoa remains controversial, as there are no universally accepted criteria that could be adopted to determine whether cocoa of a given origin is classified as fine. Therefore, the search for physical and chemical protocols to assess cocoa quality and flavor is a market demand that can be met through scientific research. The outcome may be innovation in the development of prototypes for measuring the quality of cocoa used by producers or buyers, as well as the development of new devices for cocoa fermentation and new methods of analysis, among others.

Consumer demand for higher-quality products is rising [2], which will incentivize cocoa producers to produce higher-quality cocoa. This could be a defining factor for many cocoa-producing countries, where the traditional practice has been to increase production volume to secure an income for the farmers. The need to produce a higher-quality cocoa bean should lead not only to the implementation and dissemination of the best practices in cocoa fermentation among farmers but also to the development of new fermentation methods. In this context, the use of starter cultures, tropical fruits, or immobilized enzymes to improve the sensorial perceptions of cocoa is promising, but only if the producers' incomes are high enough to cover the eventual costs associated with these practices.

## 7. Considerations and Future Perspectives

This study analyzes the cocoa fermentation research content in patent documents and scientific articles. Over the past two decades, interest in fostering R&D in cocoa fermentation has grown. The increase in scientific research by universities will stimulate innovation and technological development; hence, it is expected that universities and research institutes will assume a greater role in the development of technologies. The lead in scientific research taken by cocoa-producing countries such as Brazil, Indonesia, and Colombia over the last five years should encourage them to file for patent documents to protect their new findings, despite the barriers faced by developing countries regarding technology ownership. Therefore, this sector presents many opportunities for companies and universities in the coming years.

The technological and scientific prospects compiled in the current analysis indicate that there is a growing interest in increasing cocoa quality, which is completely aligned with

the global cocoa agreement and with the creation of cocoa quality initiatives worldwide. To achieve quality, the use of starter cultures during fermentation is the most frequent strategy used by patent applicants and researchers. The addition of exogenous materials such as fruits during fermentation is claimed by patent documents but is not frequently investigated in scientific articles, demonstrating a potential for further studies. Besides quality, there is an interest in extracting cocoa's bioactive compounds, such as polyphenols offering a high antioxidant potential. Furthermore, this study can serve as a starting point for companies and researchers who aim to develop or improve this technology in different areas, especially in the food sector.

**Author Contributions:** Conceptualization, C.O.d.S., C.D.F.R., A.T.F. and L.L.N.; methodology, C.O.d.S., C.D.F.R., A.T.F. and L.L.N.; software, L.L.N., L.d.S.F., L.S.d.A., B.L.d.M.P. and M.S.P.; valida-tion, C.O.d.S., C.D.F.R., A.T.F. and L.L.N.; formal analysis, L.L.N.; investigation, L.L.N.; resources, C.O.d.S., C.D.F.R. and A.T.F.; data cura-tion, C.O.d.S., C.D.F.R., A.T.F. and L.L.N.; writing—original draft preparation, L.L.N. and M.S.P.; writing—review and editing, C.O.d.S., C.D.F.R., A.T.F. and L.L.N.; visualization, C.O.d.S., C.D.F.R., A.T.F. and L.L.N.; supervision, C.O.d.S., C.D.F.R. and A.T.F.; project administration, C.O.d.S., C.D.F.R. and A.T.F.; funding acquisition, A.T.F. All authors have read and agreed to the published version of the manuscript.

**Funding:** The authors thank the Coordenação de Aperfeiçoamento de Pessoal de Nível Superior—CAPES (CAPES—88887.848918/2023-0); the Programa de Desenvolvimento da Pós-Graduação (PDPG)– Emergencial de Consolidação Estratégica dos Programas de Pós-Graduação stricto sensu acadêmicos (process number: 88881.708195/2022-01); and the Conselho Nacional de Desenvolvimento Científico e Tecnológico (CNPq-405707/2023-1) for financial support.

**Data Availability Statement:** Not applicable.

**Acknowledgments:** The authors are grateful to the Conselho Nacional de Desenvolvimento Científico e Tecnológico (CNPq) and the Coordenação de Aperfeiçoamento de Pessoal de Nível Superior (CAPES) for providing grants, fellowships, and financial support.

**Conflicts of Interest:** The authors declare no conflicts of interest.

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
