# Peer review of "Innovation in Cocoa Fermentation: Evidence from Patent Documents and Scientific Articles"

_fermentation, doi:10.3390/fermentation10050251_

Round 1

Reviewer 1 Report

Comments and Suggestions for Authors

This manuscript aims to analyze the technological and scientific applications regarding cocoa fermentation through the study of patent documents and research articles. The purpose of the review was to provide a technological and scientific perspective on patent documents and scientific articles regarding cocoa fermentation, through consolidated data. Brazil, Colombia and Indonesia were the countries with the highest number of scientific publications in the last 5 years, despite the lack of patents filed. United Kingdom, France, China, Canada and Germany are the main representatives in the technological field. The technological and scientific perspectives gathered in this analysis indicate that there is growing interest in increasing cocoa quality, in line with cocoa quality initiatives around the world.

The article is very interesting and pleasant to read as it is clear and well written. For publication I suggest only two small changes.

Add reference lines 50-56

Change subparagraph in 3.2 (line 157)

Author Response

We would like to thank the suggestions and criticism which were fundamental to improve the quality of this manuscript.  The paper has been rewritten in some parts aiming at complying with the suggestions of the referee.

The authors are grateful for the words and comments.

  • Add reference lines 50-56

References 3, 4, and 5 support the information (lines 50-56 – original paper). These references are mentioned in lines 50-59 of the revised version.

  • Change subparagraph in 3.2 (line 157)

The term “3.1” has been rephased with “3.2” (line 157).

Again, we would like to thank the referees for all the suggestions given, which we consider were of great value.

 Sincerely yours,

Authors

Reviewer 2 Report

Comments and Suggestions for Authors

This is an interesting manuscript dealing with the topic of inventory and analysis of the technological and scientific applications regarding cocoa fermentation through a prospective study of patent documents and research articles. The topic addressed in the manuscript is interesting, and the subject of the study is of considerable scientific and technological interest and deserves a more in-depth dissertation.

However, some aspects are recommended to be analyzed by authors and the opportunity to be taken into consideration to be assessed:

-row 186 – a mention is made about a panel, however it is not clear what authors wish to address very clearly (even the reference is clearly indicated for this paragraph).

-row 195 – please verify if the figure 1 is correctly mentioned here.

-row 202 – by “do not own the cocoa fermentation technologies” do you mean the patented cocoa fermentation technologies?

-all Latin names of microorganisms and plants should be written using Italic characters.

-as a suggestion only, the name of the sub-chapter 3 could be rephrased, in order to avoid confusion with sub-chapters 4 and 5.

-number 3.1 is assigned to two sub-chapters.

Author Response

We would like to thank the suggestions and criticism which were fundamental to improve the quality of this manuscript.  The paper has been rewritten in some parts aiming at complying with the suggestions of the referee.

  • row 186 – a mention is made about a panel, however it is not clear what authors wish to address very clearly (even the reference is clearly indicated for this paragraph).

The authors thank the observation. row 186 – the paragraph has been partially rephrased to make it clearer (lines 185-199).

  • row 195 – please verify if the figure 1 is correctly mentioned here.

The authors thank the observation. “Figure 1” has been rephased into “Figure 2” (row 199).

  • row 202 – by “do not own the cocoa fermentation technologies” do you mean the patented cocoa fermentation technologies?

“do not own the cocoa fermentation technologies” has been rephrased into “do not own patented cocoa fermentation technologies” (row 203)

  • all Latin names of microorganisms and plants should be written using Italic characters.

The authors thank the observation. All Latin names of microorganisms and plants are now written using Italic characters (as example: line 271).

  • as a suggestion only, the name of the sub-chapter 3 could be rephrased, in order to avoid confusion with sub-chapters 4 and 5.

In response to the suggestion, the title of topic 3 was complemented: 3. Technological and Scientific prospection: timeline and territoriality (row 111)

  • number 3.1 is assigned to two sub-chapters.

The topic 3.1 has been rephased into 3.2 (line 157).

Again, we would like to thank the referees for all the suggestions given, which we consider were of great value.

Sincerely yours,

Authors